# Invisible Yet Detected: PelFANet with Attention-Guided Anatomical Fusion for Pelvic Fracture Diagnosis

**Abstract.** Pelvic fractures pose significant diagnostic challenges, particularly in cases where fracture signs are subtle or invisible on standard radiographs. To address this, we introduce PelFANet, a dual-stream attention network that fuses raw pelvic X-rays with segmented bone images to improve fracture classification. The network employs Fused Attention Blocks (FABlocks) to iteratively exchange and refine features from both inputs, capturing global context and localized anatomical detail. Trained in a two-stage pipeline with a segmentation-guided approach, PelFANet demonstrates superior performance over conventional methods. On the AMERI dataset, it achieves 88.68% accuracy and 0.9334 AUC on visible fractures, while generalizing effectively to invisible fracture cases with 82.29% accuracy and 0.8688 AUC, despite not being trained on them. These results highlight the clinical potential of anatomy-aware dual-input architectures for robust fracture detection, especially in scenarios with subtle radiographic presentations.

**Keywords:** PelFANet, Invisible Fracture Detection, Anatomy-Guided Attention, Pelvic X-ray Classification

## 1 Introduction

Pelvic fractures are among the most critical injuries in emergency medicine, typically caused by high-energy trauma such as motor vehicle accidents or falls [1]. Due to the pelvis's anatomical complexity and its role in protecting vital organs and blood vessels, such fractures can lead to severe complications, including hemorrhage and multi-organ damage [2]. In-hospital mortality rates range from 5% to 20%, influenced by fracture severity, hemorrhagic shock, and associated injuries [3, 4].

Diagnosis relies heavily on radiographic evaluation and clinician expertise [5], which poses challenges in trauma settings. Subtle or complex fractures are often missed, even by skilled radiologists [6]. In high-pressure environments, diagnostic errors are common, with up to 20% of pelvic fractures initially overlooked in trauma centers [7], resulting in delayed treatment, worsening injuries, and increased mortality [8]. Rapid, accurate detection is thus essential.

Recent studies have demonstrated strong performance in pelvic and femur fracture detection using deep learning frameworks, achieving accuracies in the range of 80–98% [9, 10]. Segmentation-guided classification is a powerful method that enhances classification accuracy by localizing specific regions of interest before feature

extraction and prediction. In the context of medical imaging, this technique is especially useful for focusing on diagnostically relevant anatomical structures while ignoring irrelevant background noise. Segmentation-guided classification has proven effective across various domains, including colorectal cancer, liver cancer, and pneumonia, by improving diagnostic focus and reducing false negatives [11-15]. These methods are particularly valuable in low-contrast or cluttered imaging scenarios common issues in pelvic X-rays where global analysis may be insufficient for accurate fracture detection.

Recent studies have shown the effectiveness of segmentation-guided pelvic fracture classification. [16] reported 96.32% DSC and 98.03% accuracy using Swin U-Net, while [17] achieved 0.96–0.97 DSC and 69–88% classification accuracy across pelvic ring fracture types using an AO/OTA-guided system.

However, fracture diagnosis in pelvic radiographs can benefit significantly from contextual background information beyond the bone boundaries. While some fractures show clear cortical disruptions, others present subtle signs such as abnormal alignment, joint spacing, or limb asymmetry indicators that may lie outside the segmented bone region. Studies have shown that non-local cues like limb rotation, joint dislocation, or pubic symphysis widening can suggest fractures even in the absence of visible cortical breaks [18, 19]. Segmenting out only the bone often removes these diagnostic cues, whereas raw pelvic X-rays preserve the full anatomical context, including soft tissue and alignment, which can be crucial for detecting such subtle injuries.

This is particularly relevant in cases of invisible fractures (IV), PXR images without obvious fracture signs but confirmed via 3D-CT imaging. As demonstrated in recent work, these cases are challenging for existing deep learning methods [20]. Our work addresses this by leveraging both raw and segmented inputs to retain global structure and enhance diagnostic robustness.

To address the limitations of relying solely on either segmentation or raw image analysis, we propose PelFANet, a Pelvic Fused Attention Network that integrates both segmented bone structures and raw pelvic X-rays through a dual-stream fusion-guided attention architecture. By combining localized anatomical detail with full-field context, PelFANet is designed to detect both overt and subtle fracture cues. The streams are fused using CBAM-based attention [21], allowing the model to learn feature combinations from both inputs that contribute to improved classification. PelFANet outperforms existing approaches by accurately detecting both visible fractures and invisible fractures, by leveraging subtle contextual and anatomical cues indicative of underlying injury.

## 2     Methodology

### 2.1    Overview

Our proposed framework for pelvic fracture classification follows a two-stage pipeline combining bone segmentation and dual-stream classification. First, a U-Net with Mix Transformer B0 encoder generates segmentation masks from raw pelvic X-rays, from which bone regions are cropped to create the bone segmentations [22, 23]. These

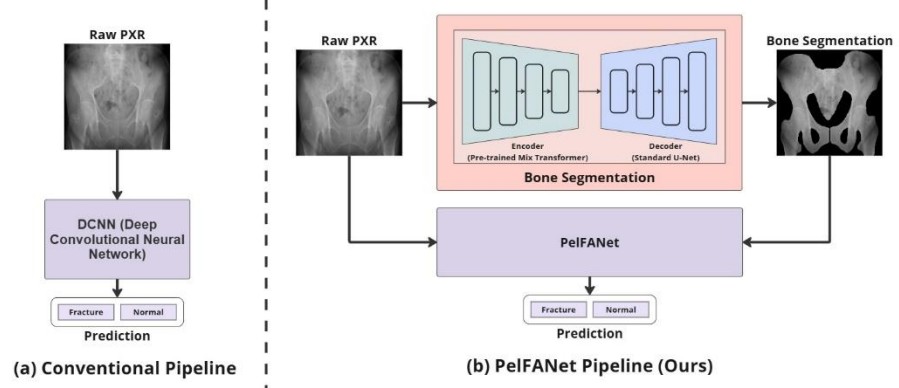

**Fig. 1.** (a) Conventional single-stream pipeline where only the raw pelvic X-ray (PXR) is used for direct fracture prediction. (b) Our proposed PelFANet pipeline where the raw PXR is first processed by a segmentation model to generate a bone segmentation. Both the raw image and the segmentation are then jointly input into PelFANet, a dual-stream fused attention network, to predict the presence of fractures.

cropped segmentations, along with the corresponding raw X-rays, are fed into PelFANet, a dual-stream network that extracts and fuses features using Fused Attention Blocks (FABlocks) with CBAM. This architecture effectively integrates local anatomical detail with global context to enhance fracture detection. The complete pipeline is illustrated in Figure 1.

### 2.2 Bone Segmentation

To incorporate anatomical context into the classification process, we first generate bone segmentations using a U-Net with a Mix Transformer B0 encoder. This hybrid architecture combines U-Net's spatial accuracy with the transformer's global context modeling, enabling precise delineation of pelvic bones.

Considering the full Pelvic region as a single class we train a one-class segmentation model. Once trained, the model infers bone masks, which are cropped to produce bone segmentations used as the second input to PelFANet, guiding fracture classification with structure-aware features.

### 2.3 PelFANet

**Input.** PelFANet uses a dual-input design, combining each raw pelvic X-ray with its corresponding bone segmentation. Both inputs are resized into a 224x224 image, then passed into two streams for separate processing. This setup enables the network to leverage both structural and contextual cues for accurate fracture classification.

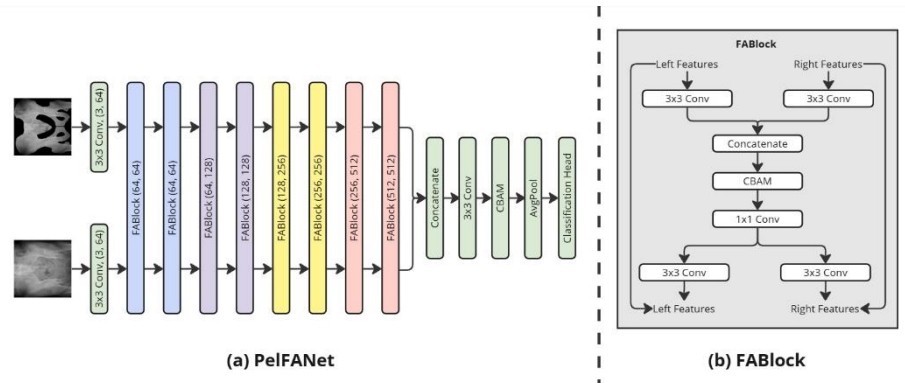

**Fig. 2.** (a) The overall architecture of PelFANet, a dual-stream fused attention network designed for pelvic fracture classification. It processes raw pelvic X-rays and corresponding bone segmentations through parallel convolutional branches, followed by stacked eight FABlocks for progressive feature fusion and attention refinement. (b) Detailed structure of a FABlock (Fused Attention Block), which extracts stream-specific features, applies CBAM-based attention over the concatenated representation, and redistributes refined features back into the two branches via residual pathways.

**Network Architecture.** PelFANet is a dual-stream convolutional architecture specifically designed to fuse global context from raw pelvic X-rays and fine-grained anatomical structure from bone segmentations. The network is composed of three main stages: parallel feature extraction, attention-guided fusion through stacked FABlocks, and final aggregation and classification.

In the initial stage, the raw pelvic X-ray and its corresponding bone segmentation are passed through two independent convolutional branches. Each stream begins with a 3×3 convolution layer, followed by batch normalization, ReLU activation, and max pooling. These parallel branches extract low-level features specific to the raw and segmented modalities.

The core of the network consists of eight Fused Attention Blocks (FABlock). At each FABlock, the feature maps from the left and right branches are independently processed, concatenated, and passed through a Convolutional Block Attention Module (CBAM) to generate attention-refined fused features. These fused features are then projected via a 1×1 convolution and split back into the two original streams. Residual connections and convolutional layers further refine the separated features before reconcatenation. This repeated fusion and redistribution mechanism allows the model to dynamically integrate complementary features across modalities while maintaining stream-specific information.

Following the FABlocks, the fused features undergo global attention refinement and pooling before final classification through a fully connected layer.

This architecture illustrated in Figure 2, enables the network to reason jointly over global cues and localized bone structures, improving its ability to detect subtle or complex fracture patterns that may not be captured by single-source models.

**FABlock.** The Fused Attention Block (FABlock) is the core unit of PelFANet, enabling interactive feature refinement between the raw pelvic X-ray stream and the bone segmentation stream. Let the input feature maps from these two streams be $F_1 \in R^{C \times H \times W}$ and $F_2 \in R^{C \times H \times W}$, respectively.

First, each input is passed through a stream-specific convolution:

$$F_1' = f_{3 \times 3}(F_1), \ F_2' = f_{3 \times 3}(F_2) \tag{1}$$

The outputs are concatenated channel-wise, where $F_{cat} \in R^{2C \times H \times W}$:

$$F_{cat} = Concat(F_1', F_2') \tag{2}$$

This combined feature map is refined using the Convolutional Block Attention Module (CBAM). CBAM sequentially applies channel and spatial attention to highlight informative features. The CBAM-refined fused feature map is denoted as:

$$CFA = f_{1 \times 1}(CBAM(F_{cat})) \tag{3}$$

This output, CFA (Combined Feature with Attention), is then used to update the original streams using additional convolution and residual addition:

$$NF_1 = F_1 + f_{3 \times 3}(CFA), \ NF_2 = F_2 + f_{3 \times 3}(CFA) \tag{4}$$

where $NF_1$ and $NF_2$ are the updated feature maps for the raw X-ray and segmentation streams, respectively. These outputs are then forwarded to the next FABlock, enabling progressive cross-stream refinement with attention.

**Final Feature Aggregation and Classification.** The fused feature map from the final FABlock is passed through a 3×3 convolution followed by CBAM attention, batch normalization, and ReLU activation. Global features are then extracted using adaptive average pooling and flattened into a 1024-dimensional vector. This vector is passed through a fully connected layer to produce the final classification output, enabling prediction of fracture presence based on the combined raw and anatomical information.

## 3    Experiments

### 3.1    Datasets

**AMERI Dataset.** The Visible Fracture subset of the AMERI PXR dataset consists of 228 pelvic X-ray images, including 168 fracture cases and 60 normal cases. These were selected from a larger set of 481 pelvic X-rays collected from 315 subjects at *** Hospital in *** between April 2013 and August 2019. All fracture cases were confirmed by experienced radiologists. To ensure data quality, cases with implants or incomplete pelvic coverage were excluded.

We also curated a dedicated Invisible Fracture subset comprising 23 fracture and 12 normal cases. These fractures are not visible in X-rays but were confirmed through

**Table 1.** PelFANet Performance on Visible and Invisible Subsets

| Fracture Type | Accuracy | Precision | Recall | Specificity | F1 Score | AUC |
|---------------|----------|-----------|--------|-------------|----------|-----|
| Visible | 0.8868 | 0.9249 | 0.9221 | 0.7833 | 0.8471 | 0.9334 |
| Invisible | 0.8229 | 0.8836 | 0.8435 | 0.7833 | 0.8123 | 0.8688 |

corresponding 3D-CT scans, providing a challenging benchmark for evaluating the model's ability to detect subtle and context-dependent fractures.

**COVID QU-Ex Dataset.** We utilize the COVID-QU-Ex dataset for pretraining, which comprises 33,920 chest X-ray (CXR) images categorized into three classes: COVID-19 (11,956), Non-COVID infections such as viral or bacterial pneumonia (11,263), and Normal (10,701) [24-28]. Crucially, the dataset provides ground-truth lung masks for all images, making it one of the largest public sets. This paired data enabled effective pretraining before fine-tuning on pelvic X-rays.

### 3.2    Segmentation Training and Setup

We trained a U-Net with Mix Transformer B0 on the AMERI dataset using 2-fold cross-validation (228 images split equally, resized to 224×224). Data augmentation included geometric (ShiftScaleRotate, Perspective, Crop, Padding), intensity (CLAHE, Brightness-Contrast, Gamma), and texture/color (Sharpening, Blurring, Motion Blur, HSV) applied probabilistically. The binary segmentation model used sigmoid activation and Dice Loss, trained for 300 epochs with Adam optimizer and Learning Rate (LR) $2\times10^{-4}$ and cosine annealing scheduler (min LR $1\times10^{-5}$, cycle 50), batch size 25.

### 3.3    PelFANet Training and Setup

The model was pretrained on the COVID-QU-Ex dataset because it exposed the network to a wide range of anatomical structures and radiographic variations, reducing the likelihood of over-specialization to the training set.

 The dataset was split into 80% training and 20% testing, with 20% of the training set used for validation. Each image was augmented four times, via random rotation (±25°), shearing (±10%), horizontal flipping, and translation (±10%) expanding the training set to 108,575 images.

 Pretraining used CrossEntropyLoss, an SGD optimizer (LR=0.0001), and a StepLR scheduler (decay every 10 epochs by 0.1). The model was trained for 100 epochs with a batch size of 64.

 For fine-tuning, 5-fold cross-validation was applied to the visible subset. To address class imbalance, each fracture case was augmented into 2 variants and each normal case into 6, using the same augmentation strategy. The final classification layer was changed from three to two outputs, and the entire model was retrained using the same loss, optimizer, and scheduler. Fine-tuning ran for 30 epochs with a batch size of 8.

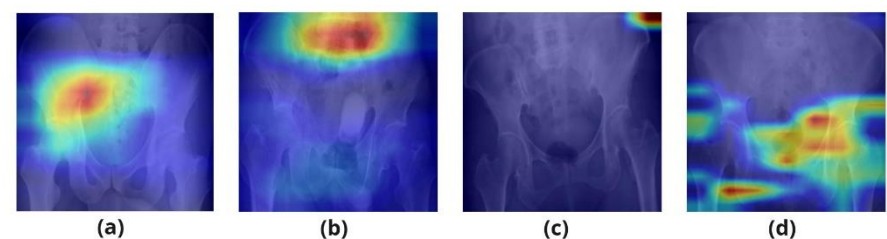

**Fig. 3.** Grad-CAM visualizations from PelFANet for different prediction scenarios. (a) Ground Truth (GT): Fracture, Predicted: Fracture correct positive detection. (b) GT: Fracture, Predicted: Normal missed fracture. (c) GT: Normal, Predicted: Normal correct negative. (d) GT: Normal, Predicted: Fracture false positive. Heatmaps indicate regions influencing the model's decision.

## 4    Result

### 4.1    Segmentation Performance

The bone segmentation model was evaluated using Intersection over Union (IoU) and F1 Score. The model achieved high segmentation accuracy across both folds.

An average IoU 0.9028 and an average F1 Score of 0.9278, these results confirm the effectiveness of our segmentation setup, providing reliable and accurate anatomical masks that serve as critical inputs to the PelFANet classifier.

### 4.2    PelFANet Classification Performance

Following segmentation, the PelFANet architecture processes both the raw PXR and the segmentation mask to perform fracture classification. Performance metrics, averaged across the 5-fold setup, are shown in Table 1. On the test set of visible fracture cases, PelFANet achieved an accuracy of 88.68%, precision of 92.49%, recall of 92.21%, and an AUC of 0.9334. While the model demonstrates strong sensitivity with high recall, the specificity of 78.33% indicates a moderate rate of false positives, reflecting a trade-off between detecting fractures and avoiding misclassification of normal cases. Most importantly, PelFANet was evaluated on the challenging invisible fracture subset, where fractures are not visible in the pelvic X-rays. Although trained exclusively on visible fracture cases, the model generalized well to this difficult set, achieving 82.29% accuracy, 88.36% precision, 84.35% recall, 78.33% specificity, and an AUC of 0.8688. These results suggest that PelFANet captures deeper, more abstract fracture features by effectively integrating both global context and localized anatomical information.

Combining both raw PXR images and bone segmentation masks with attention mechanisms likely contributed to this improved performance. The bone segmentations not only guide the model to focus on diagnostically important regions but also retain spatial correspondence with the raw input, which is especially useful for subtle or non-local signs of fracture. As illustrated in Figure 3, Grad-CAM visualizations reveal that

**Table 2.** Comparison with Prior Methods

| Method | AUC | F1 Score | IV AUC | IV F1 Score |
|---|---|---|---|---|
| ImageNet [20] | 0.8961 | 0.8000 | 0.7549 | 0.7270 |
| DRR20 [20] | 0.9290 | **0.8520** | 0.8002 | 0.7860 |
| ImageNet + DRR20 [20] | 0.9280 | 0.8390 | 0.7140 | 0.7210 |
| ImageNet + DRR20_Full [20] | 0.9151 | 0.8330 | 0.6896 | 0.7750 |
| PelFANet (Ours) | **0.9334** | 0.8471 | **0.8688** | **0.8123** |

correctly classified fracture cases exhibit focused activation on relevant bone regions, while correctly classified normal cases show minimal activation. In contrast, misclassified samples tend to display scattered or misplaced attention, reflecting uncertainty in the model's decision-making.

### 4.3    Comparison with Prior Methods

To validate the effectiveness of PelFANet, we compare it against multiple baselines from previous work that used ResNet-based classifiers [29] with various pretraining strategies, including ImageNet, DRR20 synthetic data, and their combinations. As shown in Table 2, PelFANet outperformed all prior models across both visible and invisible test sets.

On the visible test set, PelFANet achieved an AUC of 0.9334, slightly outperforming the previous best method DRR20 with an AUC of 0.929. More importantly, PelFANet showed a substantial improvement on the challenging invisible fracture subset, achieving an AUC of 0.8688 and an F1 score of 0.8123, which is significantly higher than the prior best DRR20 with an AUC of 0.8002 and F1 score of 0.786.

This comparative analysis highlights the distinct advantage of our dual-input, attention-fused framework, which enables PelFANet to capture both global context from raw images and precise anatomical boundaries from segmentations. Unlike conventional single-stream or pretraining-only methods, our architecture dynamically refines features across both modalities through FABlocks and CBAM, leading to better performance especially when facing complex or subtle fracture patterns.

## 5    Conclusion

In this study, we proposed PelFANet, a segmentation-guided dual-stream attention network designed to improve pelvic fracture classification, with a focus on invisible fractures. By integrating raw pelvic X-rays and corresponding bone segmentations, PelFANet leverages global anatomical context alongside localized structural cues. Its Fused Attention Blocks enable effective feature interaction between inputs, guiding the model to attend to diagnostically relevant regions. Results show PelFANet outperforms prior methods, especially in detecting invisible fractures, highlighting the potential of anatomy-aware dual-input models for real-world diagnostic challenges. Future work will extend this framework to other anatomical regions and larger datasets to support real-time clinical use.

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
