# OpenReview forum: "Invisible Yet Detected: PelFANet with Attention-Guided Anatomical Fusion for Pelvic Fracture Diagnosis"
_MICCAI.org/2025/Workshop/MSB_EMERGE — MSB EMERGE 2025 Conditionalrequiresmajorrevision_

### Official Review · Reviewer_gjvx · 2025-07-03

**Recommendation:** 3
**Confidence:** 3

**Clarity:**

The paper is clear and well-written, with minor areas for improvement in clarity

**Feedback:**

- The abbreviation CBAM is used first and only explained later.
- On page 5 use \mathbb{R} for real values!
- The variables C,H,W are not introduced on page 5.
- The residual connection in the FABlock is a bit hidden. Maybe you can make it more obvious by introducing a block with a + or "Add". You still have more than enough space!
- The resolution of your Figures is awful. Try to export your Figures as pdf or SVG, or at least export in high resolution!
- I recommend formatting all results (in results + astract) to the same format XX.XX%
- Some of the information in Sec. 2.1 and 2.2 are duplicated. I recommend removing details from Sec. 2.1 and adding additional details (e.g. on the UNet with Transformer) in Sec. 2.2
- CBAM is a new concept to me. Maybe you can include it in Fig. 2 underneath (a) as a third block (c).
- In Sec. 3.3 you write values in parentheses like so $(\pm20°)$. What are those supposed to mean? Give some explanation for it!
- In Sec. 3.3 you write about a visible subset. I guess you mean the subset of visible fractures? Name it better. Maybe even give the two sets names like \textit{VIS} and \textit{INVIS} subsets.
- The F1 score in Segmentation is better known as Dice score (Sec. 4.1)
- Sec. 4.1 Sentence 3 misses a verb.
- Table 1 is way too far away from the accompanying text. Place it closer!
- In Fig. 3 include the Ground truth and prediction in the image next to the label. For example: a) GT: F; Pred: NF. In the caption, write that GT stands for Groundtruth, F and NF for fracture and no fracture etc. That makes it easier to comprehend. Also, you don't need to specify false positives etc. Those are obvious given the GT and pred.
- I assume IV in Tab. 2 stands for invisible. I actually found that this abbreviation was introduced in the introduction, but was never used after. At least make it clearer in the caption of the table!
- The functions $f$ in FABlock (2.3) are not the same convolutions, are they? I would give each a separate name or include parameters like $f(F_1, \theta_1)$
- I believe in Sec. 3.1 you forgot to include the hospital name, from which the dataset originates.

**Justification:**

The experimental design is a major issue of this manuscript. My final recommendation will depend on the rebuttal.

**Reproducibility:**

Sufficient amount of details available for reproducing the main results, but open access is not provided to source code and/or data

**Strengths:**

- The paper is well written.
- The proposed architecture outperforms the baselines.
- Figures 1 and 2 look very good, accompanying the text well.

**Summary:**

The authors propose an architecture consisting of a segmentation model and a classifier that takes both the original image and the segmentation classifying whether the pelvic x-ray contains a fracture.

**Weaknesses:**

- The entire data splitting is incomprehensible to me. The pre-training dataset is split into train/val/test, but why do we have a test split, if this is just used for pre-training? The dataset for training the segmentation model is split into 2 folds. However, the PelFANet is fine-tuned on 5-folds. Why this sudden change in splitting? Do you ensure that the test data was never seen by neither the UNet nor the PelFANet?
- Have the models in Sec. 4.3 been trained on the subset of visible or invisible fractures?
- How is the PelFANet pre-trained, requiring the outputs of the UNet that was not pre-trained?
- In section 3.2, you mention the usage of Sigmoid activations. I guess this applies only to after the last layer?
- The paper misses a baseline experiment with the same pre-training data used for training PelFANet. From the presented results, I cannot conclude whether the increase in performance originates from the pre-training data or the proposed model architecture.

---

### Official Review · Reviewer_uxyS · 2025-07-07

**Recommendation:** 3
**Confidence:** 4

**Clarity:**

The paper is generally clear but has some clarity issues that could be addressed with moderate revision

**Feedback:**

Except for handling the weaknesses listed:

-   Advantage of providing the masked X-rays over the binary segmentation mask to the two
    stream classification model should be investigated.
-   Formatting: distance between text and tables seems to be too small.

**Justification:**

Lot of concepts and abbreviations not introduced. Comparison to ResNet models insufficient.

**Reproducibility:**

Not enough amount of details available for reproducing the main results, and open access details are unclear

**Strengths:**

1.  The paper is well written and relatively easy to follow with descriptive figures.
2.  The authors curate a specific dataset to train their model.
3.  Detection of pelvic fractures depicts a clinically relevant task deep learning can help to improve.

**Summary:**

The authors design a two stream model for classification of pelvic fractures taking input from raw and masked X-ray images. For automatic generation of segmentation mask a UNet model is trained, resulting in a two stage approach. The method is developed based on a private dataset curated for model training, pretraining is performed on a public COVID dataset. For comparison, the ability of a ResNet-based model using different pretraining strategies is investigated.

**Weaknesses:**

4.  There are several abbreviations used not introduced or only later on, i.e.
    - AO/OTA, page 2
    - PXR, page 2
    - CBAM, page 2
    - SGD, page 6
5.  There are several concepts/methods used not introduced, i.e.
    - Mix Transformer B0
    - Grad-CAM
    - DRR20
6.  Grad-CAM visualizations in Fig. 3. do not provide any real value.
    The underlying X-ray image is not visible and a non-expert viewer is not able to detect
    if the heat maps highlight any regions of interest, i.e. fractures, or if they are random.
7.  Comparison of ResNet model are insufficient. Reference model in Table 2. are trained on
    different datasets than the model developed. Therefore, direct comparison is insufficient
    as differences are very likely to stem from the different dataset on the model architecture.
8.  Missing reproducibilty of the U-Net model. The U-Net model gets not clear from the paper.
    Despite the fact that "Mix Transformer B0" is not introduced, it is completely unclear
    how the decoder structure looks like.

---

### Official Review · Reviewer_cv8f · 2025-07-10

**Recommendation:** 2
**Confidence:** 3

**Clarity:**

The paper has significant clarity issues that hinder understanding, substantial revision is required to improve clarity

**Feedback:**

Add important ablations. Compare against recent/stronger baselines classification models. Isolate and quantify the effect of segmentation fusion. Provide standard deviation/errors bars, not just average metrics. Improve clarity in language and technical writing (for example, "the classification layer was changed from three to two outputs” needs rephrasing). Consider qualitative analysis of model failures (for false positives/negatives). For future journal extension, increase the size and diversity of the size and diversity of the invisible dataset. Even a public release of these rare cases (under license) would be a strong contribution. In future, move towards localization and grading of fractures instead of binary classification. The authors could also explore multi-modal input (for example, clinical data) or temporal context (for example, prior imaging). The authors can also consider semi-supervised or contrastive pretraining using unlabelled pelvic X-rays instead of chest images and explore more anatomically related pretraining strategies (e.g., synthetic pelvic CT projections).

**Justification:**

While the paper addresses a meaningful clinical problem, the proposed method offers only an incremental contribution over existing segmentation-guided classification approaches. The evaluation is limited: key design choices are not ablated, the invisible fracture test set is very small, and stronger recent baselines are not considered. Metrics more appropriate for imbalanced data, like F1 or AUPRC, are underexplored. As a result, the work does not convincingly show clear technical or empirical advancement.

**Reproducibility:**

Not enough amount of details available for reproducing the main results, and open access details are unclear

**Strengths:**

Detecting pelvic fractures in a 2D radiograph (especially the invisible ones) is an important problem in trauma settings and it is often missed on radiographs. The idea is simple and intuitive: combining full raw image with segmented bone regions trying to capture both global and local information. Despite no invisible samples in training, the model performs reasonably on the invisible test set. Though minimal, the Grad-CAMs add some interpretability into model focus.

**Summary:**

The paper presents PelFANet, a dual-stream attention-based network that fuses raw pelvic X-rays with bone segmentation masks to detect pelvic fractures (particularly the "invisible" ones which show no visible signs on radiographs but are confirmed via 3D CT). The method uses a U-Net with Mix Transformer encoder for bone segmentation followed by a classification model that integrates features from raw and segmented images via Fused Attention Blocks (FABlocks) with CBAM attention. The paper claims improved results over baselines on the AMERI dataset and despite being trained only on visible fractures, the model generalizes to invisible cases.

**Weaknesses:**

- The justification for novelty is weak. The field already has segmentation-guided classifier and dual-input pipelines in multiple modalities. The paper itself reports some works obtaining 80-98% accuracy in fracture detection. The authors doesn't clearly identify what new technical or clinical gap it is solving that prior work hasn't.
- The evaluation is limited and lacks rigour.
(a) The authors don't test the effect of the key components (segmentation, CBAM, FABlocks) via ablation studies.
(b) There is no baseline for raw-only + attention. This is critical. To support the claim that segmentation fusion helps, a raw-only model with CBAM should be evaluated.
(c) The dataset size is too small. Especially with the invisible subset (23 fractures + 12 normals), it is difficult to generalize performance from this. No error bars or statistical tests are presented. Are the metrics averaged across folds? If yes, the variance isn't reported.
- COVID chest X-ray dataset is used for pretraining. This is anatomically unrelated to pelvic X-rays. It's unclear how this helps and it isn't discussed. The effect of pretraining isn't isolated in any ablation. No model trained from scratch is compared to show benefit.
- The dataset and protocol details are insufficient. It's not clear how many patients the AMERI dataset has. Was patient-level splitting ensured? Otherwise, dataset leakage is possible. Segmentation uses a 2-fold CV, but it's unclear which fold's model is selected or how folds map to classification. The process of obtaining invisible fractures (via CT confirmation) is under-explained. Were these retrospective or rendered from CT? Is this subset publicly available?
- All baselines are ResNet variants (is this pretrained on ImageNet?). There's no comparison against modern backbones (including vision transformers). Some of the cited prior works already use segmentation-guided classification and even anatomy-aware approaches, yet aren't compared here.
- Some key recent methods (e.g., Anatomy-Guided Pathology Segmentation, MICCAI 2024) are not cited or compared against.
- In fact, in F1 score, PelFANet underperforms DRR20 in the visible fracture case (0.8471 vs 0.8520), which is a major weakness since accuracy can be misleading can be misleading in imbalanced data.

---

### Decision · Program_Chairs · 2025-07-18

**Decision:**

Conditional Accept (requires major revision)

**Comment:**

The paper is conditionally accepted to the EMERGE workshop. While the reviewers recognize the value of the contribution, we ask the authors to clearly state the limitations and adjust the presentation of novelty accordingly in the camera-ready version.

Acceptance is contingent on satisfactorily addressing these concerns.